# Optimism/pessimism and associations with life event perceptions

**Marcus A. Ward**[1], **William J. Chopik**[2]*

**1** Department of Psychology, Alabama State University, Montgomery, Alabama, United States of America
**2** Department of Psychology, Michigan State University, East Lansing, Michigan, United States of America

* chopikwi@msu.edu

## Abstract

Optimism is the generalized sense that good things will happen in the future, and people higher in optimism typically experience a host of positive personal and relational outcomes. However, when ostensibly important life events happen to optimists and pessimists, they rarely change their perspective about the future. One potential reason optimists are resilient to life circumstances is that they might vary in how they perceive those circumstances. Another source of confusion is whether these perceptions are driven by optimistic thinking per se or the *lack of* pessimistic thinking. In the current study, we examined how optimists and pessimists differ in their perceptions of life events in a large sample ($N$ = 929) of college students answering questions about hypothetical life events. The pessimism scale largely drove perceptions that life events are unlikely to change someone's personality, such that the four findings from the composite scale were found for the pessimism subscale but only two were found for the optimism subscale. Nevertheless, pessimists tended to think that life events were unlikely to change their worldview, were more externally controlled, were less emotionally significant, and were more likely to negatively affect their social standing. Aside from these aggregate findings, optimism and pessimism were not systematically and consistently related to the perceptions of particular life events. These findings provide additional context for individual differences in life event perceptions and provide some future directions for why life events either do or do not motivate changes in optimism and pessimism.

## Introduction

Optimism is the generalized sense that good things will happen in the future, and people higher in optimism typically experience a host of positive personal and relational outcomes. However, when ostensibly important life events happen to optimists and pessimists, they rarely change their perspective about the future. One potential reason optimists are resilient to life circumstances is that they might vary in how they perceive those life circumstances [1,2]. However, a comprehensive assessment of how optimism and pessimism are associated with life event perceptions is unavailable, particularly how optimists/pessimists perceive different types of life events. The current study examined how optimists and pessimists differ in their perceptions of life events using a recently developed taxonomy of life event characteristics.

**Data availability statement:** All data, syntax, and materials files are available from the Open

Science Framework (DOI: 10.17605/OSF.IO/YF2SQ); also see https://osf.io/yf2sq.

**Funding:** The author(s) received no specific funding for this work.

**Competing interests:** The authors have declared that no competing interests exist.

## Optimism, pessimism, and important life outcomes

On its face, optimism is associated with ostensibly positive outcomes, both individually and interpersonally. People with higher levels of optimism live longer, have better cognitive health, are happier, have more satisfying relationships, and experience more professional success across various contexts [3–11]. The most common explanation for why optimists fare better is because they are more effective at goal setting and pursuit, have more adaptive coping strategies, and feel a sense of control over their lives [2,12–15]. Although there is literature suggesting that optimism might be associated with unrealistic perceptions of the world and that too much optimism can be a bad thing [4,16–18], most studies operationalize optimism as a protective factor or even asset for intra- and interpersonal outcomes.

Recent theorizing and research have challenged the narrative that optimism and pessimism are opposite ends of a single dimension. Although it may seem confusing at first, the idea is that there are both optimistic and pessimistic ways of thinking and that they are not necessarily always in opposition to each other. For example, people could be optimistic about some domains (or overall) but pessimistic about others [19,20]. Likewise, people might simultaneously use some optimistic thinking strategies and sometimes succumb to pessimistic thinking. One of the most common ways of assessing optimism is via The Life Orientation Test (LOT-R), which typically operationalizes the construct as a single dimension, with the high end corresponding to optimism and the low end corresponding to pessimism [15,21].

Despite its original establishment as a single dimension, there are a few reasons to think that optimism and pessimism might be separable [22–25]. For example, factor analyses of the LOT-R reveal a two-factor solution—denoting optimism and pessimism as separate constructs. Also, optimism and pessimism change differently across the lifespan, and both have distinct heritable and environmental influences, as seen in twin studies [26–33]. Critics of this point will often point to the fact that the factors are perfectly aligned with the valence of the items—positively worded items form an optimism factor and negatively worded items form a pessimism factor [24,34].

Advocates for the two-factor approach to optimism will also point to discriminant validity in its relationships with health and well-being. For instance, optimism is generally linked with positive health behaviors, better physical health, and greater psychological well-being [3]. In contrast, pessimism is typically associated with more deleterious health outcomes, higher levels of stress, and poorer mental health. One of the most comprehensive (and direct) comparisons of whether optimism and pessimism predict different outcomes is a meta-analysis of over 200,000 participants by Scheier and colleagues [22] showing that the predictive power of pessimism predicting worse health is nearly double that of optimism. The authors of that particular effort point to some of the same evidence for distinguishing between optimism and pessimism—factor analyses, behavioral genetic studies, differences in coping styles, and differential associations with important outcomes. However, they also made a more straightforward observation: just because someone isn't optimistic does not necessarily mean they are a pessimist. Likewise, just because someone isn't a pessimist does not make them an optimist. However, fewer studies have focused on the antecedents to these outcomes—how people perceive and respond to stressful life circumstances when they happen (which are thought to put into motion positive and deleterious effects on health and well-being). In the current study, we focused on how optimists and pessimists might differ in their perceptions of life events.

## Optimism, pessimism, and how people change in response to life events

Personality is thought to be both stable and has the capacity to change across the lifespan [35–37]. Theories explaining why personality changes are numerous, implicating intrinsic,

volitional, biological, and environmental processes [38,39]. On average, people tend to change in ways that suggest greater maturity (i.e., more conscientiousness and agreeableness, less neuroticism; [40,41]). Many environmental explanations posit that some external shock or exogenous force spurs personality change among individuals. Life events are one of the most prominent exogenous forces studied to date. However, evidence that life events change personality has been underwhelming. In a scoping review of life events and personality change, Bleidorn and colleagues [42] report that oftentimes the same life event is associated with varied results—sometimes being associated with trait increases, decreases, or no change at all (with varying effect sizes, too).

The same equivocal findings are also seen for optimism. Namely, there are relatively consistent increases in optimism across life until older adulthood, after which it declines [32,43–45]. The mechanisms behind such increases are thought to be corresponding shifts towards positivity and emotional balance [46] and gains in accomplishments and autonomy (i.e., that good things [and people's ability to pursue them] accrue over time; [47]). Nevertheless, important life events that likely implicate these mechanisms have little effect on changes in optimism, although the few studies on this topic occasionally find some life events do so [4,32,43]. This may seem surprising—if optimism is the expectation that good things happen in the future, shouldn't optimism increase when ostensibly good things happen to people? It does not appear to be the case.

But why has the literature on life events and changes in psychological traits been so underwhelming? To date, one critique of the life events literature has been that researchers have operationalized "life events" too narrowly and decontextualized—testing whether the mere presence or absence of an event is associated with psychological change. But, of course, life events are multifaceted, and people perceive them differently. For instance, one person might view a divorce positively and represent the end of an acrimonious relationship. Another person might view it negatively and as unexpected or a source of personal failure. Yet, traditionally, both of these people's experiences have been grouped. Recently, researchers have developed a taxonomy for characterizing variations in these life event perceptions, assessed through the Event Characteristics Questionnaire [48]. For this particular taxonomy, there are nine dimensions on which perceptions of life events vary: valence, challenge, extraordinariness, impact, external control, social status change, emotional significance, predictability, and the capacity to change a person's worldview. Since its inception, the taxonomy has dramatically expanded our understanding of individual differences in life event perceptions, how perceptions are related to changes in personality and well-being, and associations with mental health [49–52].

Although there has been work linking some psychological characteristics to life event perceptions (i.e., Big Five personality traits; [53]), there have not been many comprehensive assessments of how optimism is associated with life event perceptions. What is currently known about optimists' perceptions of life events is relatively piecemeal. For example, one of the most prominent findings is that optimists tend to think that they have *control* over their life outcomes [2,54,55]. This perceived control over (positive) life events and circumstances (and avoidance of responsibility for negative life events/circumstances) is thought to be one of the reasons why optimism is associated with so many good outcomes [56–59]. Optimists also tend to view positive life events as more *impactful* (in that they think a single, positive life event will affect multiple areas of their lives and will continue to do so in the future; [2,3,54]). Finally, inherent in the definition of optimism is an assumption that they can emphasize the positive aspects of an experience (and expect more of those positive aspects in the future) [15,60]. As a result, with respect to evaluating the *valence* of a life event, it is reasonable to expect that optimists might view life events as more positively tinged than pessimists (and that

they perceive negative events in a less negative light). Indeed, when life events are perceived as more positive, they have the potential to engender optimism [43]. However, beyond perceptions of control, impact, and valence, a comprehensive assessment of optimism/pessimism-related differences across various perceptual dimensions has not been conducted. Knowing whether optimists and pessimists perceive events differently in terms of the challenges they provide, how much they might affect their social status, and whether they think their life circumstances are stable and predictable would likely provide additional information for why there are disparate outcomes between the two groups and maybe even why optimism changes (or doesn't) in response to life events.

## The current study

In the current study, we examined whether optimists and pessimists differed in how they perceived various life events. We asked about the nine perceptual dimensions from the life event characteristics taxonomy with the Event Characteristics Questionnaire [48] and whether they thought particular life events might have the potential to change their personalities [53]. Replicating past research, we hypothesized that optimists would likely perceive positive life events as internally caused/controllable, more impactful, and more positively valenced (and negative life events as externally caused, less impactful, and less negatively valenced). Associations with other perceived event characteristics were treated as exploratory. We examined these associations first by using the traditional unidimensional optimism measure. However, given research suggesting that optimism and pessimism may be separable constructs [22], we also examined whether the optimism and pessimism subcomponents would have differential associations with life event perceptions.

## Method

Our study design followed that of Rakhshani and colleagues [53] closely but focused on the study of optimism/pessimism. Data, syntax, and study materials can be found at the OSF site for this project (https://osf.io/yf2sq). The study was not pre-registered. This study was carried out in accordance with the recommendations of Michigan State University's Institutional Review Board (IRB# x16-1291e) and run online with informed consent being secured from all participants (by clicking a next arrow; documentation requirement waived). Data were analyzed anonymously.

### Participants and procedure

Participants were 929 undergraduate students who participated in the research study in exchange for course credit. Data were collected from March 27, 2023 through August 14, 2023. They ranged in age from 18 to 35 ($M_{age}$ = 19.73, $SD$ = 1.55) and were mostly women (74.3%) followed by men (24.6%) and non-binary (1.1%). The sample was mostly White (68.4%), followed by Asian (13.1%), Black (6.6%), multi-racial (4.6%), Hispanic/Latino (3.4%), and 3.9% other races/ethnicities.

Following the procedure by Rakhshani and colleagues [53], participants first filled out a series of individual difference measures (including optimism) and other measures not pertinent to the current report. Each participant completed a measure of life events, marking whether they experienced each event (and if so, when). Then, they randomly received two events from the broader list that they did not experience (see Table 1 for a full list) and rated their perceptions of those events based on event characteristics (see below).

Like Rakhshani et al., participants reported how likely each life event might change their personalities and indicated if they experienced a particular life event themselves (and if so,

**Table 1. Life Events, Perceptions of Change, and Associations with Optimism/Pessimism.**

| Life Event | M | SD | Correlations | | | | | |
| --- | --- | --- | --- | --- | --- | --- | --- | --- |
| | | | LOT-R Optimism | | Optimism subscale | | Pessimism subscale | |
| | | | r | rp | r | rp | r | rp |
| Married | 1.042 | 0.922 | 0.018 | 0.022 | 0.009 | 0.013 | −0.022 | −0.025 |
| Divorced | 1.249 | 0.942 | 0.020 | 0.023 | −0.024 | −0.019 | −0.055 | −0.057 |
| Entered the workforce | 0.931 | 0.844 | 0.054 | 0.059 | 0.041 | 0.049 | −0.055 | −0.056 |
| Fired from job | 0.988 | 0.899 | 0.004 | 0.007 | −0.004 | −0.0002 | −0.010 | −0.012 |
| Laid off from job | 0.972 | 0.920 | 0.016 | 0.018 | 0.010 | 0.014 | −0.018 | −0.019 |
| Moved to a new city or town at least 50 miles (80km) away | 1.081 | 0.885 | 0.001 | 0.006 | −0.003 | 0.005 | −0.004 | −0.006 |
| Close friend died | 1.464 | 0.863 | 0.017 | 0.021 | −0.038 | −0.034 | −0.064 | **−0.067** |
| Romantic partner died | 1.570 | 0.883 | 0.024 | 0.028 | −0.035 | 0.031 | **−0.073** | **−0.076** |
| Father or mother died | 1.508 | 0.908 | 0.023 | 0.025 | −0.034 | −0.032 | **−0.070** | **−0.071** |
| Close family member died | 1.233 | 0.878 | 0.054 | 0.056 | 0.007 | 0.011 | **−0.085** | **−0.086** |
| Became a parent | 1.497 | 0.879 | 0.027 | 0.030 | −0.034 | −0.031 | **−0.078** | **−0.080** |
| Became seriously ill or injured | 1.303 | 0.891 | 0.043 | 0.045 | 0.011 | 0.014 | −0.063 | −0.065 |
| Jailed or imprisoned | 1.341 | 0.936 | 0.003 | 0.005 | −0.025 | −0.021 | −0.027 | −0.028 |
| Close family member jailed or imprisoned | 0.812 | 0.897 | −0.001 | 0.001 | −0.001 | 0.004 | 0.003 | 0.004 |
| Spent significant time in a different country | 0.991 | 0.874 | 0.054 | 0.063 | 0.017 | 0.027 | **−0.076** | **−0.082** |
| Made a new close friend | 0.682 | 0.801 | 0.059 | **0.066** | **0.072** | **0.081** | −0.035 | −0.039 |
| Had a falling out with a close friend | 0.759 | 0.799 | 0.029 | 0.037 | 0.057 | **0.068** | 0.001 | −0.002 |
| Started college or university | 1.101 | 0.806 | 0.046 | 0.054 | 0.049 | 0.058 | −0.033 | −0.039 |
| Graduated college or university | 0.841 | 0.867 | 0.021 | 0.026 | 0.009 | 0.015 | −0.029 | −0.031 |
| Victim of a serious crime | 1.376 | 0.899 | 0.017 | 0.018 | 0.001 | 0.004 | −0.029 | −0.029 |
| Experienced a natural disaster | 1.081 | 0.898 | −0.019 | −0.018 | −0.023 | −0.020 | 0.010 | 0.010 |
| Total/Average | 1.134 | 0.880 | 0.024 | 0.028 | 0.003 | 0.011 | −0.039 | −0.041 |

*Note.* Bold values are significant at $p < .05$. $r$ = correlation. $rp$ = partial correlation controlling for age and gender in a regression model.

when). Because of the infrequency of experiencing most of the life events, we chose to focus on perceptions of hypothetical events as it guaranteed a sufficient sample size for each analysis. This information is available in the shared data file on the OSF page.

## Measures

**Optimism/pessimism.** Optimism/pessimism was assessed with the Life Orientation Test-Revised (LOT-R). Studies have shown that the LOT-R has good reliability and validity [21,61]. The six items were: "If something can go wrong for me it will," "I'm always optimistic about my future," "In uncertain times, I usually expect the best," "Overall, I expect more good things to happen to me than bad," "I hardly ever expect things to go my way," and "I rarely count on good things happening to me." Participants are asked to rate the extent to which they agree with each item on a scale ranging from 1(*strongly disagree*) to 6(*strongly agree*).

In total, six items were used to assess LOT-R optimism ($\alpha = .84$). However, in their evaluation of the separability of optimism and pessimism, Scheier and colleagues [22] noted that pessimism (i.e., the negatively valenced items) often has more predictive validity than optimism (i.e., the positively valenced items). Thus, we also computed separate 3-item subscales for optimism ($\alpha = .78$) and pessimism ($\alpha = .84$).

**Beliefs about event-related personality change.** Participants rated how much they thought each of the 21 life events would change someone's personality (defined as based on ways of thinking, feelings, and behaving). Events were rated on a 5-point scale: -2(*definitely*

*no*), -1(*probably no*), 0(*maybe*), 1(*probably yes*), and 2(*definitely yes*). The full list of life events (and descriptive statistics) can be found in Table 1.

### Event perceptions

Participants also completed an 18-item version of the Event Characteristics Questionnaire for two hypothetical life events they were randomly assigned to evaluate [48]. The 18-item version provides two items to measure each of the nine dimensions: challenge ($\alpha = .83$; e.g., "This event was stressful."), change in world views ($\alpha=.59$; e.g., "This event helped me gain new perspectives."), emotional significance ($\alpha = .73$; e.g., "This event moved me a lot."), external control ($\alpha = .71$; e.g., "This event was in the hands of other people."), extraordinariness ($\alpha = .74$; e.g., "Most people like me experience this event sometime in their lives." [reversed]), impact ($\alpha = .61$; e.g., "This event had a strong impact on my life."), predictability ($\alpha = .87$; e.g., "This event occurred suddenly."), social status change ($\alpha=.76$; e.g., "My reputation suffered from this event."), and valence ($\alpha = .85$; e.g., "This event was joyful."). All items were rated using a 5-point scale ranging from 1(*strongly disagree*) to 5(*strongly agree*). For interpretation, the characteristics of control, extraordinariness, predictability, and social status were scored such that higher scores correspond to events that are more externally (v. internally) controlled, more (v. less) extraordinary, less (v. more) predictable, and more (v. less) damaging to their social status, respectively.

Worth noting is one life event was "started college or university." However, because the participants were all college students, this event was not rated on event characteristics (because they only rated two randomly chosen events that they had not experienced before).

### Analytic approach

For each analysis, we ran the analyses for the LOT-R optimism composite, optimism subscale, and pessimism subscale separately. We first began by correlating these optimism/pessimism indicators with perceptions of how much each life event would change someone's personality.

We then correlated the optimism/pessimism indicators by overall life event perceptions collapsing across life events (e.g., is optimism associated with perceiving life events to be more positive or emotionally significant?). Finally, we examined associations between optimism/pessimism indicators and perceptions by each life event separately to see whether they varied by a particular life event (instead of collapsing across all of them). When discussing the associations for the overall LOT-R scale and its subscales, we often talk about patterns in terms of people being higher or lower in optimism or pessimism. In this way, we are merely describing patterns of people who are higher or lower in optimism rather than formally defining groups of people who are higher or lower (e.g., those who are one standard deviation above or below the mean). The correlations provide an effect size metric for the magnitude of these associations.

## Results

### Perceptions of life events spurring personality change

As seen in Table 1, the mean was above zero for perceptions of personality change after each life event. This suggests that people believe these events can potentially change a person's personality. The three most significant life events were having a romantic partner die, having a parent die, and becoming a parent. The three least significant life events were making a new close friend, having a falling out with a friend, and having a close family member jailed or imprisoned (although the means for these were still above the midpoint of zero).

There were no significant correlations with the overall LOT-R optimism scale. However, pessimism was associated with a lower belief that a person's personality would change after having romantic partners die, a parent die, close family members die, and becoming a parent. People higher in optimism thought that making a new close friend would change their personality.

We also ran the aforementioned analyses in a linear regression controlling for age and gender. As seen in the partial correlations, these results were mostly consistent with the bivariate correlations with three exceptions (which were mostly in the same direction and magnitude as the bivariate correlations). Higher levels of LOT-R optimism were associated with believing that making a new close friend would change a person's personality. Higher levels of optimism were associated with thinking that having a falling out with a friend would change someone's personality. Finally, people higher in pessimism thought that having a close friend die would likely change their personality.

## Optimism/pessimism and event characteristics

As seen in Table 2, across all pooled life events, people higher in the LOT-R optimism composite tended to view life events as more likely to change their worldview, more emotionally significant, thought they had more (internal) control over them, and thought they would be less likely to result in a change in social status. The pessimism subscale was driving most of these results as pessimists thought that life events were less likely to alter their worldviews, would be less emotionally significant, more controlled by others, and more damaging to their social status. Some of these patterns were consistent with the optimism subscale analysis (i.e., worldview, emotional significance), but not others. The lone association specific to the optimism subscale was that people higher in optimism thought that life events were more ordinary.

Controlling for age and gender yielded nearly identical results, with one exception. In the bivariate correlations, higher levels of LOT-R optimism were associated with perceiving more control over a life event ($r = -.046$, $p = .049$). However, this correlation became marginally significant after controlling for age and gender ($r_p = -.043$, $p = .068$).0

**Table 2. Event Characteristic Descriptives and Correlations with Optimism/pessimism.**

| | | | Correlations | | | | | |
| | | | LOT-R Optimism | | Optimism subscale | | Pessimism subscale | |
| Event Characteristic Dimension | M | SD | r | rp | r | rp | r | rp |
|---|---|---|---|---|---|---|---|---|
| Challenge | 4.034 | 0.942 | −0.003 | −0.004 | −0.024 | −0.022 | −0.017 | −0.013 |
| Worldview | 3.815 | 0.837 | **0.072** | **0.074** | **0.074** | **0.081** | **−0.055** | **−0.054** |
| Emotional Significance | 4.077 | 0.817 | **0.062** | **0.062** | **0.055** | **0.058** | **−0.056** | **−0.054** |
| Control | 2.796 | 1.052 | **−0.046** | −0.043 | 0.009 | 0.012 | **0.087** | **0.084** |
| Extraordinariness | 2.556 | 1.001 | −0.016 | −0.016 | **−0.063** | **−0.067** | −0.030 | −0.033 |
| Impact | 3.994 | 0.848 | 0.009 | 0.008 | −0.006 | −0.004 | −0.023 | −0.018 |
| Predictability | 3.322 | 1.151 | 0.007 | 0.008 | 0.026 | 0.029 | 0.014 | 0.015 |
| Social Status | 2.770 | 1.127 | **−0.070** | **−0.067** | 0.009 | 0.012 | **0.127** | **0.126** |
| Valence | 2.407 | 1.319 | −0.018 | −0.015 | 0.005 | 0.007 | 0.035 | 0.031 |

*Note*. Bold values are significant at $p < .05$. r = correlation. rp = partial correlation controlling for age and gender in a regression model.

### Optimism and event characteristics for specific life events

Our previous analysis collapsed across all the rated life events to provide a broad overview of how optimists and pessimists perceive life events. However, doing so likely obscures how optimists and pessimists vary in how they perceive *particular life* events. To more succinctly summarize the results, we divided the 20 life events into ostensibly positive and negative life events (see 53, for a discussion of this classification). Due to the number of event perception dimensions (n = 9), optimism/pessimism scales (n = 3, including the LOT-R composite), and life events (n = 20; 540 total effects), these results are presented in the supplementary materials. We summarize the results below, separated by positive and negative events and presented by event characteristic dimensions (also see S1 Table). As a reminder, the event "started college or university" was not rated as a hypothetical event because all participants were college students already.

### Positive life events

For positive life events, associations with optimism and pessimism were generally inconsistent and often non-significant (S1 Fig). However, a few patterns emerged. Pessimists tended to view certain events as more challenging, externally controlled, emotionally less significant, unpredictable, and socially detrimental. For instance, pessimists found making a new friend and graduating from college more challenging, while optimists viewed getting married as less challenging. When considering control, optimists were more likely to see moving far away and making a new friend as internally driven, but this was largely influenced by pessimists perceiving these events as externally controlled.

Differences also appeared in emotional significance and valence. Optimists viewed entering the workforce as more emotionally significant, whereas pessimists placed less emotional weight on graduating from college. Pessimists were more likely to see making a new friend as a negative experience, contributing to the association between optimism and more positive perceptions of this event.

There were a few notable judgments of ordinariness and predictability. Optimists found marriage to be a routine event, while pessimists considered making a new friend more extraordinary. Pessimists also believed moving far away was unpredictable. In terms of broader impact, pessimists felt that living in a different country would not significantly affect their lives, but they thought moving far away and making a new friend could harm their social standing.

Overall, while pessimism drove many of these associations, the specific events and dimensions implicated varied, resulting in limited consistency across findings.

### Negative life events

Similarly, for negative life events, associations with optimism and pessimism were generally weak and inconsistent (S2 Fig), with most effects being non-significant or close to zero. However, a few notable patterns emerged. Pessimists tended to rate negative events as more ordinary, externally caused, predictable, and socially damaging, while also assigning less emotional impact and lower potential for changing their worldview. For example, pessimists found being victimized by a crime to be more ordinary and believed that family members or themselves being jailed was predictable. Conversely, optimists associated divorce with unpredictability.

In terms of challenge and stress, pessimists viewed being victimized as less challenging, while the death of a friend was rated as more challenging—contributing to the association between optimism and perceiving victimization as stressful but the loss of a friend as less so. A surprising effect emerged regarding control: optimists were more likely to view the death of a

parent as internally caused, a pattern mirrored by the subscales, though in opposing directions (with pessimists leaning toward external attributions).

Pessimists generally thought negative events—such as a family member being jailed or a partner dying—would have less impact on their lives, while optimists were more likely to believe that divorce could shift their worldview. Social implications also reflected this divide, as pessimists were more inclined to think that falling out with a friend, being victimized, or experiencing a natural disaster would harm their social standing. Optimists, by contrast, viewed such events as less socially compromising, largely driven by pessimists anticipating greater harm.

Interestingly, pessimists rated being victimized or having a family member jailed as more positive than less pessimistic people. However, emotional significance was largely unrelated to optimism or pessimism across most events.

Overall, pessimism was more consistently linked to evaluations of negative life events, shaping perceptions of challenge, ordinariness, external causality, predictability, and social risk. Nevertheless, the specific events tied to these patterns varied, leading to limited consistency across dimensions.

## Discussion

In this study, we examined how optimism/pessimism was associated with perceptions of life events as assessed with the Event Characteristics Questionnaire. Pessimists tended to report that life events would unlikely change their personalities. People higher in LOT-R optimism reported that life events would be more likely to change their worldview, be emotionally significant, be more internally controllable, and be less likely to negatively affect their social status. All of these associations in the general index were found for the pessimism subscale. Two of these four were also seen in the optimism subscale, and another association (that optimism was associated with thinking life events were more ordinary) was only seen in the subscale analysis. Perceptions of particular life events were more varied. Insights from the current study may prove helpful in characterizing why optimism and pessimism change in response to life events.

### Optimism, pessimism, and general perceptions of life events

Optimism is associated with many positive outcomes across various domains, including mental and physical health, close relationships, and professional settings [3–8]. Optimists' higher levels of perceived control over their lives, goal setting and pursuit, and coping and emotion regulation are thought to be the reasons why optimists fare better with these outcomes. The belief that good things will happen in the future motivates people to take actions that advance well-being. Recent work suggests that one of the more popular optimism measures (the LOT-R) can be disaggregated into optimism and pessimism subscales [22–24]. Proponents of this disaggregation suggest that doing so can deepen our understanding of how thinking about the future eventually shapes that future. Specifically, researchers have pointed out pessimism is more closely associated with important outcomes than optimism, particularly physical health outcomes [22]. Additional evidence for this distinction comes from studies of behavioral genetics, coping strategies, and measurement work, although it is still a debated topic in the literature [23,24,27,28].

In the current study, we examined associations between life event perceptions and different operationalizations of optimism (i.e., using the overall LOT-R scale and the optimism/pessimism subscales separately). Indeed, people scoring higher in LOT-R optimism tended to perceive life events in ostensibly more adaptive ways. They thought life events had the potential

to change their perspective on the world, were emotionally significant, were controllable, and were not caustic to their social status. Although controllability perceptions have been seen in past work [2,54,55], our work contributes to the literature by showing additional perceptual differences among optimists and pessimists. Although the overall LOT-R optimism scale indeed showed some correlations with life event perceptions, the subscales of optimism and pessimism demonstrated different patterns of associations. This was also true when evaluating whether or not they thought a life event could potentially change a person's personality. For example, when evaluating whether or not life events had the potential to change a person's personality, pessimists tended to give negative endorsements to every life event. In contrast, optimism was largely unrelated to beliefs about personality change. Pessimists' disbelief in whether life events would change their personalities was seen across both ostensibly positive *and* negative events. This suggests that pessimists might consider personality largely resilient to exogenous shocks like life events. Pessimists tend to have lower perceptions of self-worth and hope for change, so these resilience beliefs might stem from an overall negative perception that their personalities are immutably problematic [2,15,21,54]. Indeed, pessimists tend to hold fixed mindsets about all sorts of personal characteristics [2,62].

Although pessimists tended to believe that life events were unlikely to change a person's personality, they nevertheless perceived life events in more negative ways, including how they might impact their lives. Specifically, the associations with the overall scale with respect to how optimists perceive events as more controllable and less damaging to their social status were primarily driven by pessimists perceiving events as *less* controllable and *more damaging* to their social status. The remaining significant correlations with the LOT-R (worldview changes, emotional significance) were also explained when considering the optimism and pessimism subscales separately. Thus, the current study provides some evidence that disaggregating optimism and pessimism within the broader scale might be useful in some circumstances; in other circumstances, doing so largely reproduced the results from the total LOT-R optimism scale.

The differentiation between optimism and pessimism aligns with prior research suggesting that these traits can exist independently within individuals [15,29]. Optimist's associations with viewing life events as more ordinary, emotionally significant, and likely to change one's worldview reflect a proactive and positive engagement with life. In contrast, pessimism's associations with seeing events as more controlled by others and likely to negatively impact social status reflect a more passive and negative stance. This distinction is crucial for understanding how individuals process and respond to life events. Interventions to enhance well-being may benefit from focusing on increasing optimism and reducing pessimism separately. For instance, fostering a sense of control and emotional significance in life events may help reduce pessimistic views, while encouraging the recognition of the potential for positive change may help to foster optimism in individuals.

## Optimism, pessimism, and specific life event perceptions

Although personality changes and develops across the lifespan [63], efforts to link psychological changes to discrete life events have provided little clarity on the mechanisms behind personality change [42]. The same can be said for the development of optimism specifically—optimism and pessimism, although showing developmental changes across the lifespan, appear resilient to life events, at least how they are currently operationalized [4,32,44]. In recent years, researchers have tried to advance the study of life events by developing organized taxonomies for characterizing them and linking them to psychological traits and changes over time [48–53].

In the current study, we found that optimism and pessimism are associated with perceptions derived from this taxonomy, at least when life events are aggregated together (see previous section). However, we thought it would also be useful to examine how optimism and pessimism were associated with perceptions of particular life events. Knowing how optimists and pessimists perceive and experience specific life events provides both more precise predictions for future research and possible therapeutic and professional guidance (e.g., gives some indications for how clients react when confronted with important life transitions). On the one hand, we found that pessimism often guided perceptions of particular life events—both positive and negative. Specifically, across the various life events and among the many dimensions, pessimists tended to think that positive events were more challenging, more externally controlled, extraordinary, unpredictable, detrimental to their social standing, and negatively valenced. They thought that negative events were more ordinary, more common, and would negatively affect their social standing. Although some of these perceptions are consistent with the aggregated analyses, there were few consistent patterns across life events. In other words, there were very few significant associations between optimism/pessimism, and the few that were significant did not fall along clear-cut lines (e.g., optimists/pessimists did not differ so dramatically in how they perceived social life events or work-life events). In this way, the perceptions of particular life events align well with the lack of consistent changes seen when optimism/pessimism are measured before-to-after many of these same life events [32]. In other words, there are some more general tendencies with which optimists and pessimists view life events on the whole, but making strong distinctions between one specific life event (i.e., divorce) and another (i.e., marriage) was less fruitful in the current study.

## Limitations and future directions

The current study had many strengths. In addition to examining associations with optimism and pessimism, we also examined perceptions of a wide range of life events using a broader taxonomy. Doing so provides a comprehensive understanding of how individuals perceive these events. Nevertheless, some limitations should be acknowledged.

First, our study relied on cross-sectional data to try to make inferences about how optimism and pessimism might change in response to hypothetical life events (using participants' perceptions). Ideally, longitudinal studies would capture repeated measures of optimism/pessimism, life events as they occur, and some subjective evaluation of how those life events affect people [49,51]. Further, although we asked participants about general personality changes following life events, an expansive approach in which they are asked about particular characteristics and traits (or even the direction [i.e., positive or negative] of these changes) would have also been desirable. Integrating these expectations in the context of longitudinal data, before and after an event is experienced, would be a major advancement in this area of research [64]. Having this type of design would allow researchers to test theories of lifespan optimism changes and the impact of exogenous shocks on psychological change.

Second, our results are limited in how much they can be generalized [65]. Specifically, our sample was comprised entirely of college students. College students represent a distinct demographic group with unique characteristics and experiences. Because college students are predominantly young, it is unclear whether we would also see these same results among people over the age of 35 (the oldest person in our sample). Indeed, event perceptions tend to vary across life [66]. Because we relied on a college sample that was primarily comprised of psychology majors, our sample was also mainly comprised of women—another limitation. As a result, our findings might not fully capture the diversity of responses to life events across different demographic or cultural backgrounds. Of course, there is work examining

cross-cultural variation in optimism and future orientations more generally [67–73]. However, many of these studies have been mono-cultural or have compared a relatively limited number of cultures simultaneously. Expanding the research to include a more diverse range of participants from various demographic and cultural backgrounds is essential to understanding how optimism and pessimism are related to life event perceptions.

## Conclusion

In closure, we examined associations between optimism, pessimism, and life event perceptions. We found that pessimism drove many perceptions—pessimists thought that life events were more externally controlled, damaging to their social status, and less likely to change their worldviews or be emotionally significant. As an extension, they largely thought that life events would be unlikely to change their personalities. Although pessimism was associated with various perceptions of particular life events, these associations were few and far between, with little consistency emerging across events. Examining these questions in a longitudinal and more diverse context can advance our understanding of how life events might potentially change optimism and pessimism. Having a firmer grasp of these phenomena can also potentially help patients in therapeutic settings as they navigate important and potentially transformative life events.

## Supporting information

**S1 Table. Associations between Optimism and Pessimism and Event Characteristic Perceptions of Life Events.** Note. Correlations r ≥ |.205| are significant at p ≤ .05. Green cells are more positive; red cells are more negative; orange and yellow cells are in-between in terms of magnitude.
(DOCX)

**S1 Fig. Associations between optimism and pessimism and event characteristic perceptions of positive life events.** Next two pages. Black marker: LOT-R optimism; red marker: optimism subscale; blue marker: pessimism subscale.
(DOCX)

**S2 Fig. Associations between optimism and pessimism and event characteristic perceptions of negative life events.** Next five pages. Black marker: LOT-R optimism; red marker: optimism subscale; blue marker: pessimism subscale.
(DOCX)

## Author contributions

**Conceptualization:** William J. Chopik.

**Data curation:** William J. Chopik.

**Formal analysis:** Marcus A. Ward, William J. Chopik.

**Investigation:** William J. Chopik.

**Methodology:** William J. Chopik.

**Project administration:** William J. Chopik.

**Resources:** William J. Chopik.

**Supervision:** William J. Chopik.

**Visualization:** William J. Chopik.

**Writing – original draft:** Marcus A. Ward.

**Writing – review & editing:** Marcus A. Ward, William J. Chopik.

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
