## [Decision Letter · Decision Letter 0]

22 Nov 2024

PONE-D-24-34519Optimism/pessimism and Associations with Life Event PerceptionsPLOS ONE

Dear Dr. Chopik,

Thank you for submitting your manuscript to PLOS ONE. After careful consideration, we feel that it has merit but does not fully meet PLOS ONE’s publication criteria as it currently stands. Therefore, we invite you to submit a revised version of the manuscript that addresses the points raised during the review process.

We look forward to receiving your revised manuscript.

Kind regards,

Mosi Rosenboim

Academic Editor

PLOS ONE

Journal Requirements:

2. We note that you have referenced (Oh J, Purol MF, Weidmann R, Chopik WJ, Kim ES, Baranski E, et al. Health and well-being consequences of optimism across 25 years in the Rochester Adult Longitudinal Study. Manuscript in preparation. 2021) which has currently not yet been accepted for publication.

Please remove this from your References and amend this to state in the body of your manuscript: (Oh J, Purol MF, Weidmann R, Chopik WJ, Kim ES, Baranski E, et al. Health and well-being consequences of optimism across 25 years in the Rochester Adult Longitudinal Study. Manuscript in preparation. 2021.) as detailed online in our guide for authors

Additional Editor Comments:

I have carefully reviewed the manuscript and the reviewers' comments. Reviewer #1 has raised serious concerns about the work. Reviewer #2 takes a more lenient view. I suggest that you address the reviewers' comments and respond to the reservations they raised regarding the manuscript.

Reviewers' comments:

Reviewer's Responses to Questions

**Comments to the Author**

1. Is the manuscript technically sound, and do the data support the conclusions?

Reviewer #1: Yes

Reviewer #2: Yes

2. Has the statistical analysis been performed appropriately and rigorously?

Reviewer #1: Yes

Reviewer #2: Yes

3. Have the authors made all data underlying the findings in their manuscript fully available?

Reviewer #1: Yes

Reviewer #2: Yes

4. Is the manuscript presented in an intelligible fashion and written in standard English?

Reviewer #1: Yes

Reviewer #2: Yes

5. Review Comments to the Author

Reviewer #1: This well-written paper studies how optimists and pessimists perceive life events differently, particularly in terms of personality change and other characteristics. The study involved 929 college students who answered questions about hypothetical life events. Pessimists tend to believe life events are less likely to change their personality, are more externally controlled, less emotionally significant, and are more likely to negatively affect social standing. Optimists view life events as more likely to change their worldview and be emotionally substantial.

The topic of the paper and the results of the study are interesting. However, I have several reservations about this paper, which are summarized as follows:

Major comments:

1) The results presented in Supplementary Figures S1 and S2 in the last section (Optimism and Event Characteristics for Specific Life Events) are difficult to follow. It would be helpful to compile the results in a table or graph to aid better understanding and comparison. Additionally, it was challenging to follow the detailed findings for Positive Life Events and Negative Life Events.

2) The analysis approach relies solely on correlations, which are quite weak. Correlation does not imply causality; it only indicates association. Thus, relying solely on correlations poses a problem. Therefore, I recommend conducting linear regression analysis. This would help in identifying causal relationships and provide more insight into the relationship between optimists/pessimists and life events, while controlling for other variables such as gender and age. I believe that incorporating regression analysis will enhance the paper and be more suitable for PLOS ONE readers.

Minor comments:

1) Line 46: The sentence starting with "one potential reason…" is unclear. Please cite the source for this determination.

2) Line 180: Most of the participants are women. The authors should explain the reason for this imbalance and consider it in the analysis, such as in the regression analysis mentioned above

3) Line 188: the sentence "that they did not experience themselves" in the parentheses should be outside the parentheses.

4) Line 197: Please add the initials for the "Life Orientation Test-Revised" as mentioned in the next sentence. I couldn't locate the six items on this scale. Please specify where it can be found.

5) Line 223-225: It should be presented as a footnote.

6) Line 250: According to Table 1, the life event "Made a new close friend" is in bold for optimism, not the life event mentioned in the text (becoming a parent).

7) Section "Optimism and Event Characteristics" starting from line 252: you explain the meaning of the correlations in Table 2. The explanation of three of them is a little bit confusing, as follows:

a. People higher in the LOT-R optimism thought they had more control over them – in the table, the correlation is negative. You should rewrite it so that the reader can understand better, or it should be better detailed in the measures section.

b. Pessimists thought that life events were…… more likely to negatively affect their social status – again, in the table the correlation is positive. Please rewrite to prevent confusion.

c. People higher in optimism thought that the life events were more extraordinary – in the table the correlation is negative. Please rewrite to prevent confusion.

8) I recommend mentioning again that the "Graduated college or university" item was omitted, possibly in a footnote.

9) In the discussion section, at line 350, you mentioned that many of the associations reported in the general LOT-R optimism scale were driven by the pessimism subscale. However, I believe it would be more accurate to say that all the associations in the general index are found in the pessimists, while two out of four also appear in the optimists. In two out of four cases, the positive direction of the relationship comes from the optimists and not from the pessimists.

10) Limitations: The ones you mentioned are valid, but please consider that the study only includes young people up to 35 years old. The results could vary for older individuals.

Reviewer #2: Thank you for the opportunity to review your manuscript, “Optimism/pessimism and Associations with Life Event Perceptions”. This manuscript offers helpful descriptive-level information on optimism, pessimism, and characteristics of life events that, as far as I know, is not reported in any other literature to-date. Overall, I thought that the manuscript was clearly-written. However, I believe there are opportunities for clarification and improvement, which I have listed below.

Largest concern:

Predicted personality trait change

o One analysis the authors complete involves participants’ predictions of personality change. However, these results are only briefly integrated into the discussion. This particular research question feels a little disjointed/removed from the rest of the study, which has a clearer focus on perceptions of life events in optimists/pessimists. I suggest that the authors better integrate personality change prediction into the discussion (maybe it would be easiest to integrate the personality-change-prediction results when discussing pessimism as the driving force behind so many of the significant associations?)

o It looks like the authors assessed personality change belief broadly (i.e., not picking out specific traits or assessing the direction of believed change). I think that this level of information has important implications for optimism/pessimism that would better flesh out this research question and tie it into the rest of the manuscript (e.g., optimists may be more likely to believe that a life event, even a negative one, could change them in positive ways). Simply out of curiosity, did you happen to collect any data that could speak to that?

Extremely picky side note: I tried to check the survey for this information, but the QSF file on OSF was giving me a hard time. Could you include a PDF (or even a .docx) of the survey, as well (for those who don’t have an application to open QSF files)?

Minor concerns:

• I found this sentence difficult to follow: “For example, proponents of distinguishing between optimism and pessimism often point to factor analyses of the LOT-R revealing a two-factor solution, the fact that they change in different ways across the lifespan and that each of these factors has distinct heritable and environmental influences, in the context of twin studies”.

• The methods section was the first place you mention the life event characteristics scale by name. For readers who might not immediately make the connection, it would be helpful to name the scale when you first describe it in the introduction, again when you reference it in “The Current Study”, and throughout.

• When unpacking your results, you compare results for “optimists” vs. “pessimists”, but don’t clearly state your metric for defining these things. Are these participants who were simply higher in one than the other, 1 SD higher/lower in optimism vs. pessimism, etc.?

• Results: I occasionally was confused about the “less/more” comparisons when breaking down the results for positive/negative life events.

• E.g., in the section “LOT-R optimism was associated with thinking that being victimized was more challenging/stressful but having a friend die was less challenging/stressful. These associations were driven by pessimists rating victimization as less challenging and friends dying as more challenging (although optimists also thought having a friend die would be less challenging).”

• I was constantly circling back to think about “pessimists think the life event was less challenging than optimists? Optimists thinking a friend’s death is less challenging than victimization? Or less challenging than pessimists?” I think this warrants some repetition for the sake of clarity.

• I think it’s confusing (and potentially misleading) to break apart the discussion into a section on ‘life outcomes” and a section on “life event perceptions”. The current study doesn’t examine life outcomes, and the information in that section seemed to be about perceptions.

• As a final thought, it might be helpful to quantify how many of the results (in a count or percentage) you believe were driven my pessimism (as that was my biggest takeaway from the paper).

6. PLOS authors have the option to publish the peer review history of their article (what does this mean? ). If published, this will include your full peer review and any attached files.

**Do you want your identity to be public for this peer review?** For information about this choice, including consent withdrawal, please see our Privacy Policy .

Reviewer #1: No

Reviewer #2: No

---

## [Author Response · Author response to Decision Letter 1]

3 Jan 2025

See attached document for appropriate formatting.

Response to Editor and Reviewers

We would like to express our gratitude to the editor and reviewers for their valuable and insightful comments on the manuscript. We greatly appreciate the constructive feedback, which has contributed significantly to the improvement of the manuscript. Below, we outline how each reviewer comment was addressed and detail the corresponding changes made to the manuscript. The reviewers' comments are presented in regular text, and our responses are highlighted in bold. We also note the line numbers where these changes were made (these line numbers correspond to the clean version of the document).

Editor

Thank you for providing this formatting guidance. We have now revised the manuscript (primarily headings needed to be revised), supplementary materials and naming conventions therein, and title page in accordance with these guidelines. We also made some minor grammatical and wording changes throughout after a graduate student provided very useful edits that enhanced the clarity.

2. We note that you have referenced (Oh J, Purol MF, Weidmann R, Chopik WJ, Kim ES, Baranski E, et al. Health and well-being consequences of optimism across 25 years in the Rochester Adult Longitudinal Study. Manuscript in preparation. 2021) which has currently not yet been accepted for publication.

Please remove this from your References and amend this to state in the body of your manuscript: (Oh J, Purol MF, Weidmann R, Chopik WJ, Kim ES, Baranski E, et al. Health and well-being consequences of optimism across 25 years in the Rochester Adult Longitudinal Study. Manuscript in preparation. 2021.) as detailed online in our guide for authors

The Oh et al. paper has subsequently been published in 2022, and we used an out-of-date citation. We have now updated the reference to the published version in the manuscript.

Reviewer #1

This well-written paper studies how optimists and pessimists perceive life events differently, particularly in terms of personality change and other characteristics. The study involved 929 college students who answered questions about hypothetical life events. Pessimists tend to believe life events are less likely to change their personality, are more externally controlled, less emotionally significant, and are more likely to negatively affect social standing. Optimists view life events as more likely to change their worldview and be emotionally substantial.

The topic of the paper and the results of the study are interesting. However, I have several reservations about this paper, which are summarized as follows:

Major comments:

1) The results presented in Supplementary Figures S1 and S2 in the last section (Optimism and Event Characteristics for Specific Life Events) are difficult to follow. It would be helpful to compile the results in a table or graph to aid better understanding and comparison. Additionally, it was challenging to follow the detailed findings for Positive Life Events and Negative Life Events.

Thank you for your positive evaluation of our manuscript. We have now created a table that summarizes the many correlations in this supplementary analysis (see S1 Table). We also dramatically simplified the description of the results in these two sections. Our main goal was to convey that there were few consistent associations found. These new sections read as follows (see lns 308-357):

“Positive life events

For positive life events, associations with optimism and pessimism were generally inconsistent and often non-significant (S1 Fig). However, a few patterns emerged. Pessimists tended to view certain events as more challenging, externally controlled, emotionally less significant, unpredictable, and socially detrimental. For instance, pessimists found making a new friend and graduating from college more challenging, while optimists viewed getting married as less challenging. When considering control, optimists were more likely to see moving far away and making a new friend as internally driven, but this was largely influenced by pessimists perceiving these events as externally controlled.

Differences also appeared in emotional significance and valence. Optimists viewed entering the workforce as more emotionally significant, whereas pessimists placed less emotional weight on graduating from college. Pessimists were more likely to see making a new friend as a negative experience, contributing to the association between optimism and more positive perceptions of this event.

There were a few notable judgments of ordinariness and predictability. Optimists found marriage to be a routine event, while pessimists considered making a new friend more extraordinary. Pessimists also believed moving far away was unpredictable. In terms of broader impact, pessimists felt that living in a different country would not significantly affect their lives, but they thought moving far away and making a new friend could harm their social standing.

Overall, while pessimism drove many of these associations, the specific events and dimensions implicated varied, resulting in limited consistency across findings.

Negative life events

Similarly, for negative life events, associations with optimism and pessimism were generally weak and inconsistent (S2 Fig), with most effects being non-significant or close to zero. However, a few notable patterns emerged. Pessimists tended to rate negative events as more ordinary, externally caused, predictable, and socially damaging, while also assigning less emotional impact and lower potential for changing their worldview. For example, pessimists found being victimized by a crime to be more ordinary and believed that family members or themselves being jailed was predictable. Conversely, optimists associated divorce with unpredictability.

In terms of challenge and stress, pessimists viewed being victimized as less challenging, while the death of a friend was rated as more challenging—contributing to the association between optimism and perceiving victimization as stressful but the loss of a friend as less so. A surprising effect emerged regarding control: optimists were more likely to view the death of a parent as internally caused, a pattern mirrored by the subscales, though in opposing directions (with pessimists leaning toward external attributions).

Pessimists generally thought negative events—such as a family member being jailed or a partner dying—would have less impact on their lives, while optimists were more likely to believe that divorce could shift their worldview. Social implications also reflected this divide, as pessimists were more inclined to think that falling out with a friend, being victimized, or experiencing a natural disaster would harm their social standing. Optimists, by contrast, viewed such events as less socially compromising, largely driven by pessimists anticipating greater harm.

Interestingly, pessimists rated being victimized or having a family member jailed as more positive than less pessimistic people. However, emotional significance was largely unrelated to optimism or pessimism across most events.

Overall, pessimism was more consistently linked to evaluations of negative life events, shaping perceptions of challenge, ordinariness, external causality, predictability, and social risk. Nevertheless, the specific events tied to these patterns varied, leading to limited consistency across dimensions.”

2) The analysis approach relies solely on correlations, which are quite weak. Correlation does not imply causality; it only indicates association. Thus, relying solely on correlations poses a problem. Therefore, I recommend conducting linear regression analysis. This would help in identifying causal relationships and provide more insight into the relationship between optimists/pessimists and life events, while controlling for other variables such as gender and age. I believe that incorporating regression analysis will enhance the paper and be more suitable for PLOS ONE readers.

We appreciate the suggestion to re-run the analyses controlling for age and gender. We now report these results in the form of partial correlations (which control for age and gender, derived from the context of a regression analysis) in Tables 1 and 2. We also updated the Results section to describe any differences between the models.

Minor comments:

1) Line 46: The sentence starting with "one potential reason…" is unclear. Please cite the source for this determination.

We have now provided citations to Scheier et al., 1986 and Peterson, 2000 as support for this claim (see ln 55).

2) Line 180: Most of the participants are women. The authors should explain the reason for this imbalance and consider it in the analysis, such as in the regression analysis mentioned above

As recommended, we have now conducted linear regression analyses controlling for gender (and age). We have now also added the imbalance as a limitation to the paper, which reads as follows (see lns 478-479):

“Because we relied on a college sample that was primarily comprised of psychology majors, our sample was also mainly comprised of women—another limitation.”

3) Line 188: the sentence "that they did not experience themselves" in the parentheses should be outside the parentheses.

We have now corrected this error.

4) Line 197: Please add the initials for the "Life Orientation Test-Revised" as mentioned in the next sentence. I couldn't locate the six items on this scale. Please specify where it can be found.

We have now added these initials as directed. The LOT-R is a widely used scale and googling the sample item yielded several results with the full items. Nevertheless, for ease, we now report all six items in the text. That text reads as follows (lns 206-210):

“The six items were: “If something can go wrong for me it will,” “I’m always optimistic about my future,” “In uncertain times, I usually expect the best,” “Overall, I expect more good things to happen to me than bad,” “I hardly ever expect things to go my way,” and “I rarely count on good things happening to me.””

5) Line 223-225: It should be presented as a footnote.

We originally planned on moving this information to a footnote. However, PLOS ONE guidelines forbid footnotes (https://journals.plos.org/plosone/s/submission-guidelines).

6) Line 250: According to Table 1, the life event "Made a new close friend" is in bold for optimism, not the life event mentioned in the text (becoming a parent).

We thank you for pointing out this mistake. Indeed, the table was correct but the text was wrong. We’ve now fixed it accordingly.

7) Section "Optimism and Event Characteristics" starting from line 252: you explain the meaning of the correlations in Table 2. The explanation of three of them is a little bit confusing, as follows:

a. People higher in the LOT-R optimism thought they had more control over them – in the table, the correlation is negative. You should rewrite it so that the reader can understand better, or it should be better detailed in the measures section.

b. Pessimists thought that life events were…… more likely to negatively affect their social status – again, in the table the correlation is positive. Please rewrite to prevent confusion.

c. People higher in optimism thought that the life events were more extraordinary – in the table the correlation is negative. Please rewrite to prevent confusion.

We thank the reviewer for mentioning these issues. We agree that the scoring of the event characteristics measure can be a bit unintuitive. In response to this comment, we elected to clarify the scoring of the subscales in the Method section to reduce confusion, particularly those that might be confusing. This addition now reads (lns 235-239):

“For interpretation, the characteristics of control, extraordinariness, predictability, and social status are scored such that higher scores correspond to events that are more externally (v. internally) controlled, more (v. less) extraordinary, less (v. more) predictable, and more (v. less) damaging to their social status, respectively.”

We also revised some of the sentences noted by the reviewer to help with interpretation. We also revised a mistake in the direction of the extraordinariness finding. Specifically, this new section reads (lns 280-288):

“As seen in Table 2, across all pooled life events, people higher in the LOT-R optimism composite tended to view life events as more likely to change their worldview, more emotionally significant, thought they had more (internal) control over them, and thought they would be less likely to result in a change in social status. The pessimism subscale was driving most of these results as pessimists thought that life events were less likely to alter their worldviews, would be less emotionally significant, more controlled by others, and more damaging to their social status. Some of these patterns were consistent with the optimism subscale analysis (i.e., worldview, emotional significance), but not others. The lone association specific to the optimism subscale was that people higher in optimism thought that life events were more ordinary.”

8) I recommend mentioning again that the "Graduated college or university" item was omitted, possibly in a footnote.

Unfortunately, PLOS ONE forbids footnotes. However, we added this reminder on lns 304-306 when we discuss perceptions of particular life events (however, minor correction to the reviewer, the life event was “started college or university,” as everyone in the sample was already a college student).

9) In the discussion section, at line 350, you mentioned that many of the associations reported in the general LOT-R optimism scale were driven by the pessimism subscale. However, I believe it would be more accurate to say that all the associations in the general index are found in the pessimists, while two out of four also appear in the optimists. In two out of four cases, the positive direction of the relationship comes from the optimists and not from the pessimists.

We agree that this is a more accurate way of portraying the results (although only one of the associations was found among only optimists, not two). We have revised that sentence to replace it with the following (lns 364-367):

“All of these associations in the general index were found for the pessimism subscale. Two of these four were also seen in the optimism subscale, and another association (that optimism was associated with thinking life events were more ordinary) was only seen in the subscale analysis.”

10) Limitations: The ones you mentioned are valid, but please consider that the study only includes young people up to 35 years old. The results could vary for older individuals.

We agree that this is also a limitation to the current study. We have now added it to the Discussion (lns 475-478):

“Because college students are predominantly young, it is unclear whether we would also see these same results among people over the age of 35 (the oldest person in our sample). Indeed, event perceptions tend to vary across life.”

Thank you for your thoughtful review!

Reviewer #2:

Thank you for the opportunity to review your manuscript, “Optimism/pessimism and Associations with Life Event Perceptions”. This manuscript offers helpful descriptive-level information on optimism, pessimism, and characteristics of life events that, as far as I know, is not reported in any other literature to-date. Overall, I thought that the manuscript was clearly-written. However, I believe there are opportunities for clarification and improvement, which I have listed below.

Largest concern:

Predicted personality trait change

o One analysis the aut

---

## [Decision Letter · Decision Letter 1]

2 Mar 2025

Optimism/pessimism and associations with life event perceptions

PONE-D-24-34519R1

Dear Dr. Chopik,

We’re pleased to inform you that your manuscript has been judged scientifically suitable for publication and will be formally accepted for publication once it meets all outstanding technical requirements.

Kind regards,

Mosi Rosenboim

Academic Editor

PLOS ONE

Additional Editor Comments (optional):

Reviewers' comments:

Reviewer's Responses to Questions

**Comments to the Author**

1. If the authors have adequately addressed your comments raised in a previous round of review and you feel that this manuscript is now acceptable for publication, you may indicate that here to bypass the “Comments to the Author” section, enter your conflict of interest statement in the “Confidential to Editor” section, and submit your "Accept" recommendation.

Reviewer #1: (No Response)

Reviewer #2: All comments have been addressed

2. Is the manuscript technically sound, and do the data support the conclusions?

Reviewer #1: Yes

Reviewer #2: Yes

3. Has the statistical analysis been performed appropriately and rigorously?

Reviewer #1: Yes

Reviewer #2: Yes

4. Have the authors made all data underlying the findings in their manuscript fully available?

Reviewer #1: Yes

Reviewer #2: Yes

5. Is the manuscript presented in an intelligible fashion and written in standard English?

Reviewer #1: Yes

Reviewer #2: Yes

6. Review Comments to the Author

Reviewer #1: Thank you for all the revisions you made. You improved the paper. Even though I wanted a regression analysis, the partial correlation is fine as well. I enjoyed reading your manuscript.

Reviewer #2: My original review of this manuscript was largely favorable; the authors have adequately altered the manuscript to address my concerns.

7. PLOS authors have the option to publish the peer review history of their article (what does this mean? ). If published, this will include your full peer review and any attached files.

**Do you want your identity to be public for this peer review?** For information about this choice, including consent withdrawal, please see our Privacy Policy .

Reviewer #1: No

Reviewer #2: No

---

## [Editor Report · Acceptance letter]

PONE-D-24-34519R1

PLOS ONE

Dear Dr. Chopik,

I'm pleased to inform you that your manuscript has been deemed suitable for publication in PLOS ONE. Congratulations! Your manuscript is now being handed over to our production team.

Kind regards,

on behalf of

Dr. Mosi Rosenboim

Academic Editor

PLOS ONE